# Isolation and Characterization of Endophytes Bacterial Strains of *Momordica charantia* L. and Their Possible Approach in Stress Management

**DOI:** 10.3390/microorganisms10020290

**Published:** 2022-01-26

**Authors:** Ritu Singh, Kapil Deo Pandey, Monika Singh, Sandeep Kumar Singh, Abeer Hashem, Al-Bandari Fahad Al-Arjani, Elsayed Fathi Abd_Allah, Prashant Kumar Singh, Ajay Kumar

**Affiliations:** 1Department of Botany, Sunbeam Women’s College Varuna, Varanasi 221002, India; ritu_singh414@yahoo.co.in; 2Centre of Advanced Study in Botany, Banaras Hindu University, Varanasi 221005, India; kdpandey2005@gmail.com (K.D.P.); sandeepksingh015@gmail.com (S.K.S.); 3Department of Biotechnology, School of Life Sciences, Uttaranchal University, Dehradun 248007, India; monikasingh_bhu@yahoo.com; 4Botany and Microbiology Department, College of Science, King Saud University, P.O. Box 2460, Riyadh 11451, Saudi Arabia; habeer@ksu.edu.sa (A.H.); aalarjani@ksu.edu.sa (A.-B.F.A.-A.); 5Plant Production Department, College of Food and Agricultural Sciences, King Saud University, P.O. Box 2460, Riyadh 11451, Saudi Arabia; eabdallah@ksu.edu.sa; 6Department of Biotechnology, Pachhunga University College Campus, Mizoram University, Aizawl 796001, India; prashantbotbhu@gmail.com

**Keywords:** bitter gourd (*Momordica charantia* L.), endophyte, stress tolerance, substrate utilization, enzyme activity

## Abstract

In the present study, eight endophytic bacterial strains, namely *Bacillus licheniformis* R1, *Bacillus* sp. R2, *Agrobacterium tumefaciens* R6, uncultured bacterium R11, *Bacillus subtilis* RS3, *Bacillus subtilis* RS6, uncultured bacterium RS8 and *Lysinibacillus fusiformis* RS9, were isolated from the root of *Momordica charantia* L. All the strains, except R6 exhibited positive for IAA production, siderophore production, and phosphate solubilization during plant growth-promoting traits analysis. Strains invariably utilized glucose and sucrose as a carbon source during substrate utilization, while yeast extract, ammonium sulphate, ammonium chloride, glycine, glutamine, and isoleucine as nitrogen sources. In addition, Spectinomycin was found as the most effective during antibiotic sensitivity TEST, followed by Chloramphenicol, Erythromycin, Rifampicin and Kanamycin, while Polymixin B was found least effective, while strains R1, R6, and RS8 were sensitive to all the antibiotics. Strains R1 and RS6 were able to withstand tolerance up to 10% of NaCl. The strains showing resistance against broad-spectrum antibiotics, especially chloramphenicol, can be used in hospital waste management. In addition, strains with a tolerance of 10 % of NaCl can improve plant growth in the saline affected area.

## 1. Introduction

In the last few decades, endophytic microbial communities have been gaining significant attention in various fields of agriculture, healthcare, or environmental contamination management. In agriculture, beneficial endophytes are utilized as sources of biofertilizer, as biocontrol agents or stress modulators. However, the synthesis of bioactive compounds or nanoparticles makes them a practical resource in pharmaceuticals industries or ultimately in healthcare management. In addition, mitigation of toxic industrial effluents or soil contaminants using endophytes is an emerging strategy for environment management [1]. An endophyte in general, defined as a microorganism residing inside the host tissue without causing any apparent sign of infection. Nevertheless, even with the advancement of the latest omics and technologies, which explore large number of microbial communities even in a small quantity of sample tissue, but significantly fewer endophytic microbial communities have been cultured under laboratory conditions. Plants host a diverse community of microorganisms, including bacteria, fungi, and actinomycetes, as epiphytes on the surface or endophytes inside the host tissue. Fresh vegetables and fruits are the primary habitats for epiphytic and endophytic microbial communities [2].

*Momordica charantia* L., an indigenous plant belong to the family Cucurbitaceae, commonly known as bitter melon or bitter gourd and cultivated in tropical and subtropical regions of the world. Since ancient times, *M. charantia* L. has been used as vegetables and as an Ayurveda medicine to treat several human ailments in the Indian subcontinent [3]. The nutritional and chemical compositions of *M. Charentia* L., which are highly beneficial for our daily life, are described to contain crude lipid, fibre, protein, carbohydrate and various vitamins. In addition, phytochemicals, such as alkaloids, tannins, flavonoids, saponins, and glycosides, have also been reported [4,5], presence of optimum quantity of metabolites and bioactive compounds is frequently recommended as a source of antidiabetic, anticancerous, antimicrobial, antioxidant and antihelminthic compounds [6,7,8]. 

The plant’s rhizosphere region is considered the hot spot for microbial interactions due to the secretion of nutrient-rich root exudates. Some microbial strains enter the plant tissue and reside as endophytes during the interaction. Although endophytic microorganism has been frequently observed in almost all plant tissue [9], their population varies depending upon the host tissue, plant developmental stage, seasons, and the surrounding environmental conditions [10]. Bacterial endophytes are one of the most influential groups of microorganisms, which have been broadly explored in the last two decades for their use in agriculture as biofertilizers or biocontrol agents for the management of phytopathogen during pre or postharvest conditions. The endophytic microorganism is preferred over the other plant growth-promoting rhizobacteria due to better survival and adaptation against biotic and abiotic stresses [11,12,13].

The bacterial endophytes promote the growth of host plants directly via producing phytohormones, including indole-3-acetic acid (IAA), gibberellins, cytokinins, phosphate solubilization, N_2_ fixation or indirectly via the production of antibiotics, siderophores [13,14]. In addition, endophytes secrete various types of secondary metabolites volatiles that significantly control and inhibit the growth of phytopathogen [15]. Besides this, endophytes also counteract the adverse effects of abiotic stresses such as drought and salinity [16]. According to a previously published report, Proteobacteria, Actinobacteria, Firmicutes and Bacteroidetes are reported as predominant bacterial phyla, while *Pseudomonas, Bacillus*, *Pantoea*, Acinetobacteria are the most common bacterial genera reported in various plant species [2].

The present paper describes the isolation and characterization of bacterial endophytes associated with *Momordica charantia* L. through morphology, biochemical and molecular characterization. In addition, they have also evaluated plant growth promotion ability and their possible approach in stress management through describing substrate utilization patterns, antibiotic sensitivity, and salinity stress tolerance.

## 2. Materials and Methods

### 2.1. Isolation of Bacterial Endophytes 

For isolation, locally grown *Momordica charantia* L. was collected from the B.H.U., botanical garden, India. The sample tissues were thoroughly washed with running tap water to remove adhered soil particles and surface sterilized to remove any epiphytic microorganism by following slandered protocol; sample tissue firstly dipped in 70% C_2_H_5_OH for 2 min followed by 0.5% NaOCl for 3 min and again dipped in 70% C_2_H_5_OH for 30 sec. Finally, tissues were washed with double distilled water [13]. 

However, three different methods were applied during isolation 

### 2.2. Isolation from Root Bits

Surface sterilized roots were dissected into pieces of nearly 1-cm size with the help of a flame sterilized razor blade. Root pieces were placed on the Luria Broth (LB) agar plates and incubated at 28 °C till the pinhead colonies or visible growth appeared.

### 2.3. From Transverse Section of Roots

The transverse sections of surface-sterilized roots were cut with a sterilized razor in the laminar flow hood near a spirit burner, inoculated on LB agar plates, followed by incubation.

### 2.4. Isolation from the Crushed Root 

Surface sterilized root pieces (<3 mm size) were macerated with a small amount of quartz sand in a sterilized homogenizer. Tissue extract was diluted in phosphate buffer (pH 7.0, 0.01 M), further inoculated on LB agar plates, and incubated at 30 °C for 2–4 days. For the isolation, serially diluted (10^−1^, 10^−2^) macerate was inoculated on LB agar plate, and colony-forming units (CFU) were determined after 3–4 days of incubation at 28 °C.

## 3. Characterization of Bacterial Isolates

The isolated endophytic bacterial strains were characterized by evaluating colony morphology, biochemical screening and molecular 16S rRNA gene sequence analysis, following standard protocols [13,14]

### Genetic Characterization

Genomic DNA was isolated using a Genei Pure^TM^ bacterial DNA purification kit (GeNei^TM^, Bangaluru, India) following the manufacturer’s protocol. Universal eubacterial primers F-D1 5’-CCGAATTCGTCGACAACAGAGTTTGATCCTGGCTCAG-3’ and R-D1 5’-CCCGGGATCCAAGCTTAAGGAGGTGATCCAGCC-3’ [13,14] were used to amplify the 1500 bp region of 16S rRNA gene using a thermal cycler (BioRad, Hercules, CA, USA). The amplified products were resolved by agarose gel electrophoresis (1.5%) and visualized using a gel documentation system (Alfa Imager, Alfa Innotech Corporation, San Leandro, CA, USA). The amplicons were purified using a Genei Pure^TM^ quick PCR purification kit (GeNei^TM^, Bangaluru, India) and quantified at 260 nm using a spectrophotometer taking calf thymus DNA as control. The purified partial 16S r DNA amplicons were sequenced in an Applied Biosystems 3130 Genetic Analyzer (Applied Biosystems^®^, Waltham, MA, USA). The partial sequences of nucleotides were compared with available sequences from NCBI databases, and sequences showing >99% similarity were retrieved by Nucleotide Basic Local Alignment Search Tool (BLAST N) program available at the National Center for Biotechnology Information (NCBI) BLAST server (latest accessed on 18 October 2021) [13,14].

## 4. Plant Growth-Promoting (PGP) Traits Analysis

### 4.1. Production of IAA

Bacteria cultivated at 25 ± 2 °C for 48 h in the nutrient agar supplemented with 100–400 µg/mL of L-Tryptophan were harvested through centrifugation (8000 rpm, 10 min). The supernatant (2 mL) was mixed with two drops of orthophosphoric acid and 4 mL of the Salkowski reagent (50 mL of 35% perchloric acid and1 mL of 0.5 M FeCl_3_ solution) [17]. The development of the pink colour confirmed the production of IAA. 

### 4.2. Phosphate Solubilization

Pikovskaya medium was used to observe the phosphate-solubilizing ability of the endophytic bacterial isolates by the dissolution of precipitated calcium phosphate (Ca_3_ (PO_4_)_2_). In detail, the bacterial strains were aseptically inoculated in 3–4 places on the Pikovskaya medium agar plates and incubated at 30 °C for 2–3 days [17,18]. The development of clear halo zone around the inoculated bacterial strain confirmed phosphate solubilization activity. 

### 4.3. Siderophore Production

The cultured bacterial strains were spotted on the Chrome azurol S agar plate. First, the bacterial isolates were grown overnight at 30 °C in TSB medium to obtain OD_600_ 0.5. Then, a 10 μL aliquot of each bacterial culture was inoculated onto a diffusion disc placed on the CAS-Blue Agar plates, incubated for 72 h at 30 °C, and observed daily until a yellow-orange halo was seen around the colony [19]. The development of a yellow-orange hallow zone around the bacterial spot confirmed siderophore production. 

### 4.4. Carbon and Nitrogen Utilization

Carbohydrate utilization was tested by modifying Simmon’s citrate medium with the mixture of amino acids (i.e., glutamine, cysteine, methionine, alanine, and tryptophan in equal amounts), A_6_ trace element, and multivitamin. Sodium citrate was replaced with 0.2% (*w*/*v*) of different carbohydrates in the growth medium. Growth was examined after two days of incubation and compared with the negative control. Nitrogen utilization was tested by adding 0.1% (*w*/*v*) different nitrogen sources in the glucose medium. Growth was examined after incubation for two days [13].

### 4.5. Specific Growth Rate

Cells growing in exponential phase (A_600_ = 0.3–0.5) were inoculated in 50 mL fresh LB broth. Cultures were regularly shaken (120 rpm). Growth was recorded by measuring protein content (after washing the cell biomass) and observing their absorbance at 600 nm. The specific growth rate (µ) was determined by following Myers and Kartz [20].
µ = {2.303(log N_2_ − log N_1_)}/T_2_ − T_1_(1)
where N_1_ = protein concentration or absorbance at T_1_ and N_2_ = protein concentration or absorbance at T_2_.

### 4.6. Hydrolytic Enzyme Assay

The hydrolytic enzyme assay (amylase, protease, pectinase and esterase enzymes) was performed on solid media. All enzymatic activities were performed by growing the bacterial isolates in nutrient broth for 24 h at 30 °C. The isolates were streaked on a nutrient agar (NA) plate containing 0.2% soluble starch as the substrate for the amylolytic activity. After growth, cultures were treated with Lugol’s iodine and the formation of clear halos around the colony was marked as a positive indication [21]. For the proteolytic activity, isolated bacteria were inoculated on casein agar mixed with sterilized skimmed milk. The formation of clear halos around the colony in the HCl flooded plate was confirmed as proteolytic activity [22]. To determine pectin degradation capacity, the test isolates were inoculated on a pectin agar plate, and plates were covered with acetyl pyrimidium chloride 1% developer to assess pectinolytic and esterase activity. Clear halos around the colony indicated pectin degradation [23]. However, for the esterase activity, the medium containing peptone 10.0, NaCl 5.0, CaCl_2_ 2H_2_O, 0.1, and agar 18.0 (pH 7.4) was used to determine the ester hydrolytic ability of test isolates (g/l). Sterilized Tween 80 was added to the sterilized culture media at a final concentration of 1% (*v*/*v*). The precipitation of ester compounds around the colony indicated the presence of esterase enzyme [24].

### 4.7. Antibiotic Sensitivity Test

A sensitivity test was performed using antibiotic-impregnated discs (6 mm diameter). The antibiotic sensitivity of the strains was tested against Chloramphenicol, Spectinomycin, Erythromycin, Rifampicin, and Polymixin B by the Kirby Bauer disc-diffusion method [13,14,16]. Based on the inhibition zones, organisms were categorized as resistant, intermediate, or sensitive according to the DIFCO Manual, 10th edition (1984). 

### 4.8. Stress Tolerance

The selected bacterial strains were tested for their tolerance to salt stress by exposing the bacterial cells to various NaCl concentrations (1–12% of NaCl) and NaN_3_ (0.02%). The culture tubes were incubated at 30 °C for (48 h), and absorbance was recorded at 600 nm (UV/VIS Spectrophotometer 117, Systronics, India) [13,16,25].

## 5. Statistical Analysis

All the experimental data were expressed in a mean of three measurements along with the standard error. For statistical analysis, the single-factor ANOVA (analysis of variance) followed by Tukey’s multiple range test were carried out using SPSS.16 software. Differences in mean value were considered significant at *p* < 0.05. 

## 6. Result

### 6.1. Diversity of Cultivable Endophytic Bacteria

The endophytic bacteria grown by all three sampling methods (root beads, transverse section, and crushed root) showed good yield and growth on the LB agar plate. However, we considered isolates of crushed root samples due to the diverse, dispersed, and sufficient growth of the colony. A 6.2 ± 2.5 × 10^4^ CFU/gm fresh weight bacterial colony was observed. Twenty-eight endophytic bacterial strains were isolated on the basis of morphology, which were further identified as eight different strains on the basis of biochemical and 16 S rRNA gene sequence analyses.

### 6.2. Colony Morphology and Biochemical Characteristics 

All the endophytic strains were rod-shaped, except RS8, which was round in shape. During Gram’s staining, five strains were found positive (R1, R2, RS3, RS6, RS9) and three were Gram-negative (R6, R11 and RS8). All the bacterial strains were motile except RS3. Moreover, 50% of the isolates were positive for oxidase and caused the fermentation of carbohydrates. In addition, all the strains were negative for the urease test and positive for the citrate test, except strain RS3. Six strains showed starch hydrolysis and nitrate reduction. However, all the strains were positive for the catalase test. Indole test, H_2_S production, and Phenylalanine deaminase test were found negative for all the strains. The details of biochemical characterization are described in Table 1.

### 6.3. Phylogenetic Analysis

BLAST analysis of the 16S rRNA gene of the isolated endophytic strains resulted in retrieving closely related genera from the NCBI gene databank presented in Table 2.

In detail, isolated strains showed the closest similarity with four genera, including *Bacillus*, *Agrobacterium*, *Lysinibacillus,* and uncultured bacterial strains, and identified as *Bacillus licheniformis* R1, *Bacillus* sp. R2, *Agrobacterium tumefaciens* R6, uncultured bacterium R11, *Bacillus subtilis* RS3, *Bacillus subtilis* RS6, uncultured bacterium RS8, and *Lysinibacillus fusiformis* RS9.

The phylogenetic tree established with a bootstrap neighbour-joining method is demonstrated in Figure 1, based on 16S rRNA gene sequencing. Bacterial strains fell into phylogenetic division belonging to Firmicutes and Proteobacteria. However, Firmicutes was found to be the predominant endophytic bacterial phyla in our study.

### 6.4. Plant Growth Promotion (PGP) Trait Analysis

During PGP trait analysis, all the strains except R6 solubilized phosphate and produced IAA and siderophore.

### 6.5. Carbon and Nitrogen Source Utilization Pattern 

The tested endophytic strains respond differentially during substrate utilization. All the strains efficiently utilized glucose and sucrose as carbon sources. Regarding other carbon sources, sodium citrate was utilized by strains R1, R2, R6, R11 and RS8, whereas sodium acetate, sodium formate, and mannitol were utilized by strains R11 and RS8. However, only strain R11 utilized malic acid as a carbon source. Similarly during the nitrogen utilization pattern, all the isolates efficiently utilized yeast extract, ammonium sulphate, ammonium chloride, glycine, glutamine, and isoleucine as nitrogen sources, whereas only three strains, R2, R6, RS3, utilized aspartic acid and glutamic acid. The details utilization pattern are described in the Table 3.

The variation among carbon utilization patterns was more pronounced than the nitrogen utilization pattern. Cluster analysis of strains based on their carbon and nitrogen utilization patterns revealed that strains RS3 and RS6 showing affinity to *Bacillus subtilis* were placed nearby (Figure 2). In many cases, they were widely separated, indicating different physiological and biochemical characteristics in strains with similar morphology.

### 6.6. Specific Growth Rate

The isolated endophytes were efficiently grown in LB medium, and a higher specific growth rate (0.64) was observed in the strain R1, while lower were recorded in RS9 (0.1). However, the rest of the strains have a growth rate of between 0.14–0.54. The details of their specific growth have been presented in Figure 3.

### 6.7. Hydrolytic Enzyme Assay

All the bacterial isolates showed differential enzymatic activity (Figure 4). All the strains except R6 exhibited amylase activity. Similarly, except strain R11, all the seven isolates synthesized protease. Pectinase was produced by only the strains R6 and R11, while esterase was produced by strains R1, R6, R11, and RS9.

### 6.8. Antibiotic Sensitivity

The sensitivity of all eight isolates against different antibiotics in terms of the zone of inhibition is presented in Table 4. Strains R1, R6, and RS8, were sensitive to all the antibiotics. However, strains R11 was resistant to Polymixin B and erythromycin, while RS9 was resistant to erythromycin and kanamycin. In addition, strains RS3 and RS6 were identified as resistant to chloramphenicol, while R2 showed resistance to Rifampicin.

### 6.9. Stress Resistance 

All the strains profusely grew and tolerated NaCl, up to 4% of concentration, while after rising the concentration of NaCl, strains showed a differential response. Strains R1 and RS6 had grown up to a concentration of 10 % NaCl, but no strains survived at 12 % of NaCl. Similarly, strains R1, RS3 RS6 RS8 and RS9 grew at 0.02% of sodium azide (NaN_3_) concentration, but none of the strains survived at 3% of KOH. The detailed responses are presented in Table 5.

## 7. Discussion

The excellent colonizing efficacy of the endophytic strains compared to other epiphytic microorganisms makes them a valuable resource for growth modulation and disease management in plant systems [26]. In our study, a total of 6.2 ± 2.5 × 10^4^ CFU/gm fresh weight, the bacterial population were observed, which was in the range of previously reported endophytes in sugar beet [27], lemon [28], alfalfa [29] and potato [30]. 

Endophytic bacterial strains of bitter gourd root were isolated by the enrichment culture technique, one of the conventional approaches using LB agar medium. Strains were identified based on the morphological, biochemical, and physiological characteristics accompanied by 16S rRNA gene sequencing. Since species definition using rRNA sequencing is problematic, this slow evolving molecule lacks the required level of resolution to distinguish similar species [31]. Therefore, we restrained our isolates from the status of species [32] but felt confident based on morphological and biochemical patterns [33] which allow for the rank of genera, families, and order to be determined by bacteriologists [34].

Colony morphology indicated the variation among the endophytes. The observed phylogenetic breadth covered *Firmicutes*, uncultured bacteria and *β-Proteobacteria*, in which Firmicutes, especially *Bacillus* genera, was dominant in the root of *Momordica charentia* L. *Bacillus* genera are ubiquitous nature and frequently reported as an epiphyte and endophyte in various plant species, such as *Cassia tora* [14] and turmeric [16], 

The present assay system to evaluate the functions and persistence of endophytic bacterial strains showed common traits for amylase, protease, pectinase, esterase, and motility. The isolates with positive amylase activity showed starch hydrolysis on starch agar plates. Plant tissue stores starch as a food source, and the endophytes can consume these stored starch before other new colonizers appear [35]. It has been reported that *Pseudomonas stutzeri*, *B. megaterium,* and *B. licheniformis* produce extracellular amylase [36]. Hydrolytic enzymes, such as pectinases and cellulases, play a critical role in the dissolution of cell walls or plant tissue that may help enter or colonise endophytic strains in the host tissue [9,37]. In our study, strain R6 and R11, produced pectinase, however esterase was produced by the strains R1, R6, R11, and RS9. However, a significantly more minor survey on the secretion of these enzymes by endophytes has been conducted by Elbeltagy et al. [38]. *Agrobacterium* and one uncultured strain in the present study also produced pectinase acting as virulence factors for pathogenic bacteria of plants.

Further, this enzyme might be involved in the invasion of host plants by endophytes, as reported for *Azoarcus* sp. [39] and *Enterobacter asburiae* JM22 [40]. Esterase breaks down fats into fatty acids and glycerin, bringing a reversible reaction. Esterase enzyme help in producing saponin, offering a significant medicinal property to the plant. Esterase production was as also reported in the *Pseudomonas stutzeri* [41]. 

During PGP traits analysis, all the strains showed the properties of phosphate solubilization as well as indole acetic acid and siderophore production except the strain *Agrobacterium tumefaciens* R6. However, in previous studies, the author reported the plant growth-promoting activity of *A. tumefaciens.* Phosphate solubilization, IAA, and siderophore production are the most common plant growth attributes and reported by various authors in endophytic bacterial strains [16]. Therefore, strains showing PGP activities can be used as soil or plant inoculants to enhance the growth and yields of plants. In the last two decades, various microbial strains, either epiphyte or endophyte, have been frequently utilized for plant growth promotion and the management of biotic and abiotic stress [16,17]. IAA is one of the most common phytohormones responsible for plants’ growth, cell division, and tissue differentiation. The IAA produced by a microorganism similarly interacts with the plants and plays a significant role in growth promotion and mitigation against biotic and abiotic stress. Siderophore production is another important property of plant growth-promoting bacteria that improves plant growth by enhancing iron chelation properties for the presently available iron and plays a significant role in phytopathogen control [10,13].

All the isolated strains efficiently utilized glucose and sucrose as a carbon source, and yeast extract as a nitrogen source in the study. The isolates metabolized most of the nitrogenous substrate and amino acid, similar to an earlier report [16,42]. The findings suggested that bacterial community structure dominated as bacterial endophytes in the root tissues of bitter gourd utilized various complex organic carbon and nitrogen sources produced during cell metabolism [25].

During the study of specific growth rates, the highest was recorded in strain R1, while the lowest was noted in strain RS9. These results showed that *Bacillus licheniformis* strain R1 could be employed for better and more efficient biotic and abiotic stress management. It is well reported that competition for nutrients and space is one of the prime factors for managing phytopathogens after applying biocontrol agents during plant disease management [1].

Antibiotic sensitivity and resistance differ from one strain to another, even in a single genus. The sensitivity towards antibiotics largely depends upon the resistance of plasmid carrying antibiotic resistance gene (s). The result presented in our study revealed the differential response of various bacterial strains towards the six antibiotics kanamycin, Spectinomycin, erythromycin, Rifampicin, polymixin B, and chloramphenicol. All eight strains were sensitive to Spectinomycin. More than 90% of strains were sensitive to chloramphenicol, Rifampicin, and kanamycin, whereas 80% were sensitive to erythromycin. However encased of polymixin B, most strains showed an intermediary response. Little antibiotic resistance was seen in the collection as a whole and this is consistent with results obtained by Litzner et al. [42]. In the study, *Bacillus* sp. R2 showed higher sensitivity, while *Bacillus subtilis* strains RS3 and RS6 showed resistance to chloramphenicol and a differential response of different species of the same genera. Uncultured strain R11 showed resistance to Polymixin-B and Erythromycin. Strain *Lysinibacillus fusiformis* RS9 also showed resistance to Erythromycin and Kanamycin. Strain *Bacillus* sp. R2, which showed high sensitivity against chloramphenicol during the resistance against Rifampicin. Our findings suggest that the *Bacillus subtilis* strains RS3 and RS6, which showed resistance against broad-spectrum antibiotic chloramphenicol, can be an effective solution for hospital waste management [43]

In the present study, significant variation was observed for NaCl tolerance allowing for the discrimination of bacterial group adapted to salinity. Five strains, out of eight, grew at a concentration higher than 4% NaCl and could be considered as facultative haplotypes for abiotic stress tolerance, including salinity and drought [44]. All the strains tolerated 4% NaCl, but their response differed, above the concentration of 6% NaCl. However, the strains Bacillus *licheniformis* R1 and *Bacillus subtilis* RS6 can tolerate up to 10% of NaCl of concentration. In a previous study, endophytic strains of *Bacillus* showed tolerance at up to 10% of NaCl [45]. The higher tolerance level against salinity with the plant growth promotion ability of the *Bacillus* can be used as a better alternative for agriculture in salt stress affected areas. Similarly, the resistance against sodium azide also showed the possibility of isolated endophytic strains in abiotic stress tolerance.

In recent years, the latest advanced technologies or next-generation sequencing have made it possible to explore many microbial communities, but most of them cannot grow or are not cultivable in laboratory conditions. In this context, this study will provide an insight into cultivable endophytic strains of *Momordhica charentia* L., which can be used for the growth promotion and mitigation of abiotic stresses, including salinity and drought.

## 8. Conclusions

The present study briefly describes biochemical and molecular characterization of *Momordica charentia* L endophytic bacterial isolates. Most of the isolates harbored PGP traits which showed possibility to use as plant or soil inoculants to enhance growth and disease management in plants. The highest specific growth rate was found for strain R1, which presented possible use in biocontrol agents. Similarly, Spectinomycin was the most effective antibiotic, followed by chloramphenicol, erythromycin, Rifampicin, and kanamycin, while polymixin B was the least effective. Some strains (R1 and RS6) showed tolerance up to 10% of NaCl level, whereas no isolate was resistant to 3% potassium hydroxide. As evident in the stress tolerance test, four strains, R1, RS3, RS6, and RS8, exhibited growth in 0.02% sodium azide. The strains showing resistance against the broad-spectrum antibiotic chloramphenicol can be used in hospital waste management. However, strains with salinity tolerance of up to 10% of NaCl can be an effective solution for salinity stress management.

## Figures and Tables

**Figure 1 microorganisms-10-00290-f001:**
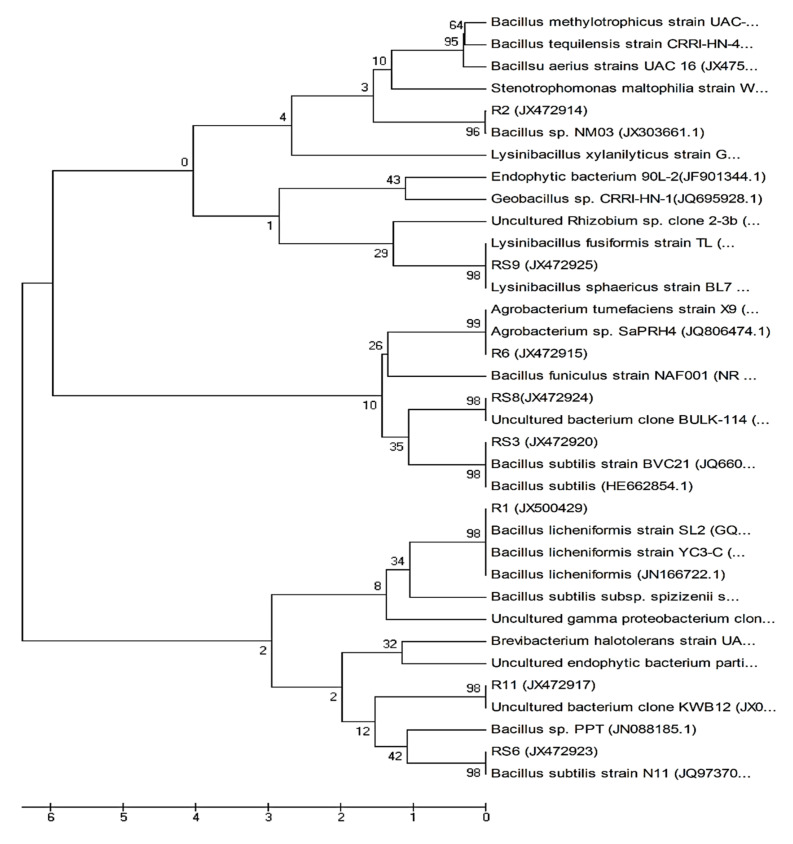
The phylogenetic tree of isolated endophytic bacterial strains was constructed using the neighbor-joining method by MEGA 5 software. Each number on a branch indicates the bootstrap confidence values corresponding to the scale bar of branch lengths.

**Figure 2 microorganisms-10-00290-f002:**
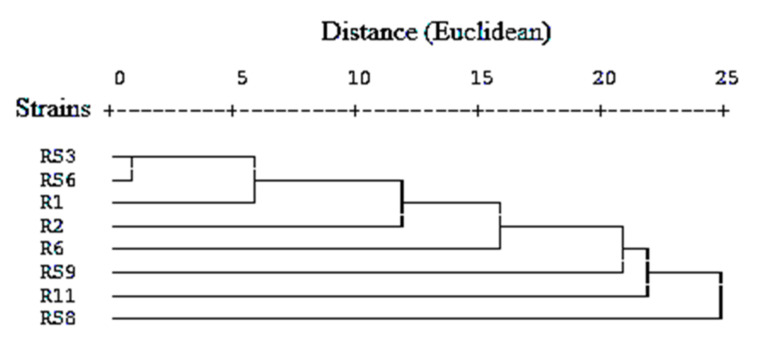
Hierarchical cluster analysis (between groups linkage) measuring the squared Euclidean distance based on their C and N utilization pattern.

**Figure 3 microorganisms-10-00290-f003:**
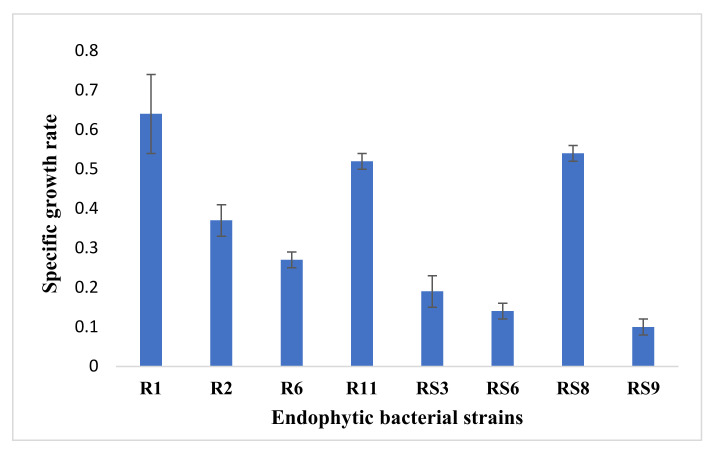
Specific growth rate (h^−1^) of the endophytic isolates grown in LB broth. The values represent ± SE.

**Figure 4 microorganisms-10-00290-f004:**
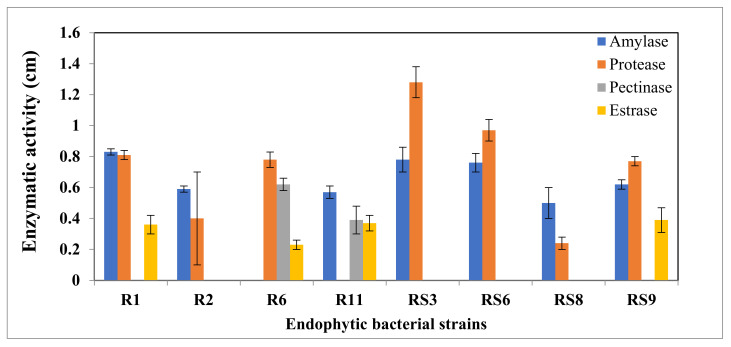
Enzymatic activity of endophytic bacterial strains, enzymatic Index represent the halo diameter of degradation/diameter.

**Table 1 microorganisms-10-00290-t001:** Biochemical characterization of endophytic bacterial strain (+ve =Positive, −ve = Negative activity).

Characteristics	R1	R2	R6	R11	RS3	RS6	RS8	RS9
Cell Shape	R	R	R	R	R	R	C	R
Gram’s Reaction	+	+	−	−	+	+	−	+
Motility	+	+	+	+	−	+	+	+
Oxidase	−	+	+	−	−	+	+	−
Citrate	+	+	+	+	−	+	+	+
Indole	−	−	−	−	−	−	−	−
H_2_S production	−	−	−	−	−	−	−	−
Phenylalanine deaminase	−	−	−	−	−	−	−	−
Carbohydrate fermentation	+	+	+	−	−	+	−	−
Starch hydrolysis	−	+	+	−	+	+	+	+
Urease	−	−	−	−	+	−	−	−
Nitrate reductase	−	+	+	+	+	+	+	−
Catalase	+	+	+	+	+	+	+	+

**Table 2 microorganisms-10-00290-t002:** The details molecular characteristics of endophytic bacterial strains.

S.N	Strains	Accession Number	Nearest Phylogenetic Neighbour	Phylogenetic Domain
1.	R1	JX500429	*Bacillus licheniformis* strain YC3-C (HQ698961.1)	Firmicutes
2.	R2	JX472914	*Bacillus* sp. NM03 (JX303661.1)	Firmicutes
3.	R6	JX472915	*Agrobacterium tumefaciens* strain X9, (JX002661)	*β-proteobacteria*
4.	R11	JX472917	Uncultured bacterium clone KWB120 (JX047140)	Uncultured bacteria clone
5.	RS3	JX472920	*Bacillus subtilis* strain BVC21 (JQ660604)	Firmicutes
6.	RS6	JX472923	*Bacillus subtilis* strain N11 (JQ973708)	Firmicutes
7.	RS8	JX472924	Uncultured bacterium clone BULK-114 (JN161941)	Uncultured bacterial strain
8.	RS9	JX472925	*Lysinibacillus fusiformis* strain TL (JQ991004)	Firmicutes

**Table 3 microorganisms-10-00290-t003:** Carbon and nitrogen source utilization patterns of endophytic bacterial strains.

Carbon Source	R1	R2	R6	R11	RS3	RS6	RS8	RS9
Glucose	+	+	+	+	+	+	+	+
Sucrose	+	+	+	+	+	+	+	+
Sodium citrate	+	+	+	+	−	−	+	−
Sodium acetate	−	−	+	−	−	−	+	−
Sodium formate	−	−	+	−	−	−	+	−
Mannitol	−	−	+	−	−	−	+	−
Malic acid	−	−	−	+	−	−	−	−
Methanol	−	−	−	+	−	−	+	+
Nitrogen Source								
Yeast extract	+	+	+	+	+	+	+	+
Potassium Nitrate	+	+	+	+	+	+	−	+
Sodium Nitrite	+	−	+	−	+	+	−	+
Ammonium acetate	+	−	+	−	+	+	+	+
Ammonium sulphate	+	+	+	+	+	+	+	+
Ammonium chloride	+	+	+	+	+	+	+	+
Alanine	+	+	+	+	+	+	+	−
Lysine	+	+	+	+	+	+	+	−
Glycine	+	+	+	+	+	+	+	+
Glutamine	+	+	+	+	+	+	+	+
Isoleucine	+	+	+	+	+	+	+	+
Arginine	−	+	+	+	+	+	+	+
Cysteine	+	+	+	−	+	+	+	+
Aspartic acid	−	+	+	−	+	−	−	−
Glutamic acid	−	+	+	−	+	−	−	−
Proline	+	+	+	+	+	+	−	−

**Table 4 microorganisms-10-00290-t004:** Antibiotic sensitivity pattern of endophytic bacterial strains (I = intermediate; S = sensitive; R = Resistance).

Strains	Antibiotic Sensitivity (Antibiotics Inhibition Zone in mm)					
	Chloramphenicol (30 mcg/disc)	Polymixin-B (300 unit/disc)	Erythromycin (15 mcg/disc)	Rifampicin (5 mcg/disc)	Spectinomycin (30 mcg/disc)	Kannamycin (10 mcg/disc)
R1	10.3 ± 0.05 ^a^(I)	14.6 ± 1.5 ^a^(I)	12.6 ± 1.5 ^a^(I)	19 ± 1 ^a^(S)	40.6 ± 3.2 ^a^(S)	23 ± 1 ^a^ (S)
R2	42.6 ± 1.15 ^b^(S)	11 ± 1 ^a^ (I)	26.3 ± 2.08 ^b^(S)	R	30 ± 1 ^b^ (S)	37 ± 2 ^b^ (S)
R6	28 ± 2 ^c^(S)	13 ± 1 ^a^(I)	19 ± 1 ^b^(S)	28 ± 2.6 ^b^(S)	23.6 ± 1.5 ^c^(S)	22.6 ± 1.5 ^a^(S)
R11	26.6 ± 1.5 ^c^(S)	R	R	10 ± 1 ^c^(I)	10 ± 1 ^d^(I)	24 ± 1 ^a^ (S)
RS3	R	12 ^a^ (I)	15 ± 1 ^a^(S)	11 ± 1 ^c^(I)	30.6 ± 1.5 ^b^(S)	30 ± 1 ^c^ (S)
RS6	R	9.6 ± 2 ^a^(I)	24 ± 1 ^b^(S)	17.6 ± 1.5 ^d^(S)	20 ± 1 ^c^(S)	26.6 ± 1.15 ^a^
RS8	39 ± 2 ^d^(S)	12.6 ± 0.5 ^a^(I)	39.73 ^c^(S)	16 ± 2 ^d^(S)	43 ± 2 ^a^(S)	40 ± 2.64 ^b^ (S)
RS9	26.3 ± 1.5 ^e^(S)	10 ± 1.54 ^a^(I)	R	14 ± 2 ^d^(I)	23 ± 1.7 ^c^(S)	R

Different letter above the data, showed a significant differences within column (*p* < 0.05).

**Table 5 microorganisms-10-00290-t005:** Response of endophytic bacterial strains against salinity and chemical stress.

Strains	concn	R1	R2	R6	R11	RS3	RS6	RS8	RS9
NaCl	4%	+	+	+	+	+	+	+	+
	6%	+	−	−	+	+	+	+	−
	8%	+	−	−	−	+	+	−	−
	10%	+	−	−	−	−	+	−	−
	12%	−	−	−	−	−	−	−	−
KOH	3%	−	−	−	−	−	−	−	−
NaN_3_	0.02%	+	−	−	−	+	+	+	−

## Data Availability

Not applicable.

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
