# Peer review of "Isolation and Characterization of Endophytes Bacterial Strains of Momordica charantia L. and Their Possible Approach in Stress Management"

_microorganisms, 2022, doi:10.3390/microorganisms10020290_

Round 1

Reviewer 1 Report

The manuscript is based on good ideas and the results recorded are useful for the promotion of sustainable agriculture in a context of intensifying climate change effects. Furthermore, the manuscript is well written, the objectives are well formulated and the overall work is well designed and sufficiently concise. As such, I strongly recommend this article to our journal. There are some minor editing suggestions attached

Author Response

 Reviewer #1

The manuscript is based on good ideas and the results recorded are useful for the promotion of sustainable agriculture in a context of intensifying climate change effects. Furthermore, the manuscript is well written, the objectives are well formulated and the overall work is well designed and sufficiently concise. As such, I strongly recommend this article to our journal. There are some minor editing suggestions below.

Response: Thanks for the comments and we are very thankful for your critical suggestion. We have modified the article as per you suggestion and  the revised text are highlighted in red

Quarry: What are the concentrations of NaCl used in your study? Only the results for the 6% to 10% NaCl concentrations have been presented in Table 5.

 Response: in the experiment, we had used 1-12 % of NaCl, we had added a separate table.6 in the manuscript for better presentation

Quarry: 4.9. Lysis in sodium dodecyl sulfate (SDS)

What is the purpose of this activity (4.9.) Lysis in sodium dodecyl sulphate (SDS) carried out during your work? The title should be reviewed and perhaps the objective of the activity should be recalled.

Response. Actually, we had used SDS to evaluate the survivality test of isolated strain against this anionic detergent, but we now removed this section from the manuscript

Quarry: The statistical analysis is not detailed. Therefore, the reference should be provided

Response. The details statistical analysis now mentioned in the revised manuscript

Quarry: 6.1. Diversity of cultivable endophytic bacteria For the Isolation, serially diluted macerate was inoculated on 3% agar with a 10% root extracts plate. LB agar plate and colony-forming units (CFU) were determined after 3-4 days of incubation at 28 °C. The above paragraph can be deleted or reworded. It looks like we are in the methodology section when we are not.

Response: Thanks for this critical suggestion, we have modified and rearrange this section in the revised manuscript.

Quarry: 8.2. Specific growth rate

Figure 3. Specific growth rate (h-1) of the endophytic isolates grown in LB broth. The values represent ±SE. the bars having different alphabets are significantly different at the level of p< 0.05.

No SE value is visible in the presented figure 3.

Response: Thanks for the comments,we have modified the figure for better visualization .

Quarry: 8.4. Antibiotic sensitivity Table 5. Antibiotic sensitivity and stress tolerance property of Momordica charantia L. endophytic strains (I- intermediate; S= sensitive; R-Resistance) +=Presence of activity, - = absence of inhibition zone;).

This table is too full. Also, some of the results for salt stress are missing. I suggest that the abbreviated names of the antibiotics used be written and that the legend be put on for the rest of the details

Response: Thanks for the suggestion, we have splitted table.5 in two for better presentation.

Quarry: The first paragraph of the discussion looks like an introduction, which is bad. The discussion is poor, could be improved.

The following paragraph should be deleted Bitter gourd plants roots were colonized internally by endophytic bacteria at an average population of 5.9 × 104 – 1.56 × 104 CFU/ g fresh root wt. in different seasons, and their population varied from 0.9 × 104 to 1.83 × 105 CFU/ g fresh root wt at different plant.

Response: Thanks for the suggestion, we have modified this paragraphs in the revised manuscript

Quarry: Update the references in the discussion. Most of your references are very old

Response. We have added the some latest references

The conclusion should be completed with an indication of the prospects for the outcome of your study

Response. The outcome of the study has been now mentioned in the conclusion section

Quarry: The abbreviated form of the journals should be presented according to the instructions to the authors. There is a need to harmonise

Rahman, M.M.; Kamrunnahar, Abu Bakar Siddique, A.B.; Bhuiyan S.R; and Zeba, N. Morphological characterization, character association and path analysis of bitter gourd (Momordica Charantia L.) genotypes. Plant Cell Biotechnology and Molecular Biology, 2021 22, 53-62

Singh, R.; Kumar, A.; Bhuvaneshwari, K.; Pandey, K.D. Gas ChromatographyMass Spectrometry Analysis and Photochemical Screening of Methanolic Fruit Extract of Momordica charantia. J. Recent Adv. Agri. 2012. 1, 122-127

Response:  Thanks for comment, we have revised the references as per your suggestion.

Reviewer 2 Report

At present, due to the enthusiasm of researchers for omix technologies, the classical study of microflora has receded into the background, although this is an important problem for the general development of plants. The resistance of plants to unfavorable external influences depends on the general composition of microorganisms. In general, there is no doubt about the results and conclusions obtained. However, the manuscript is carelessly framed and contains many misprints. There are also a number of comments:
1. Renumber sections and subsections
2. Edit the text of subsection 2.4
3. Name of subsection 3.1 -16S rRNA. rRNA and rDNA write together and further in the text check
4.Section 4.1 - write  µg/ml
5.Section 4.2 correctly write calcium phosphate
6.Section 4.3 - OD600

7.Subsection 6.1 to move to the section Materials and Methods
8. Designation of H2S productions
9. to remove table 3, does not carry information
10. repeat table 5, poorly readable text, indicate statistics
11. Plant stress tolerance is a multi-faceted problem. It is justified that the authors did not limit themselves to studying the composition of microflora, but also their resistance to certain stress factors. Unfortunately, this part of the work is shown only schematically, in table 5 it is better to indicate the % of cell death, rather than + and -.
12. Determination of extracellular enzymes is an important indicator. However, the authors determined their only qualitative availability. It is necessary to quantify the activity and present the data in the results section. It would be useful to determine the content of enzymes, both extracellular and intracellular.
13. In the discussion section, give the number of antibiotic-resistant strains, not their %

Author Response

At present, due to the enthusiasm of researchers for omix technologies, the classical study of microflora has receded into the background, although this is an important problem for the general development of plants. The resistance of plants to unfavorable external influences depends on the general composition of microorganisms. In general, there is no doubt about the results and conclusions obtained. However, the manuscript is carelessly framed and contains many misprints. There are also a number of comments:

Response: Thanks for the comments and we are very thankful for your critical suggestion. We have modified the article as per you suggestion and the revised text are highlighted in red

Quarry 1.Renumber sections and subsections

Response. Thanks for suggestion, we have revised accordingly
Quarry 2. Edit the text of subsection 2.4

Response. Done
Quarry 3. Name of subsection 3.1 -16S rRNA. rRNA and rDNA write together and further in the text check

Response. Done
Quarry 4.Section 4.1 - write  µg/ml

Response. Done
Quarry 5.Section 4.2 correctly write calcium phosphate

Response. Done
Quarry 6.Section 4.3 - OD600

Response. Done

Quarry 7. Subsection 6.1 to move to the section Materials and Methods

Response. Done, thanks for the suggestion
Quarry 8. Designation of H2S productions

Response: Thanks for the comments, we have mentioned the references which are also present in the text

Kumar, A.; Singh, R.; Yadav, A.; Giri, D.D.; Singh, P.K.; Pandey, K.D. Isolation and characterization of bacterial endophytes of Curcuma longa L. 3 Biotech. 2016. 6, 60.

Kumar, V.; Kumar, A.; Pandey, K.D.; Roy, B.K. Isolation and characterization of bacterial endophytes from the roots of Cassia tora L. Ann. Microbiol. 2015. 65, 1391-1399.

Quarry 9. to remove table 3, does not carry information

Response. Thanks for this suggestion, however, we want to retain it in text for better presentation
Quarry 10. repeat table 5, poorly readable text, indicate statistics

Response. Thanks for the suggestion, we have splitted table 5 in two for better presentation and also added the statistics

Quarry 11. Plant stress tolerance is a multi-faceted problem. It is justified that the authors did not limit themselves to studying the composition of microflora, but also their resistance to certain stress factors. Unfortunately, this part of the work is shown only schematically, in table 5 it is better to indicate the % of cell death, rather than + and -.

Response. Thanks for the suggestion, yes we are agree with your feedback. However in this experiment, we have targeted salinity stress (NaCl) or chemical stress (KOH and NaN3) for screening purpose of endophytic strains. So that, we can use it as inoculants to modulates the concentration of Momordicine (an important constituents).of  Momordica charentia L,  and the experiment is undergoing.

Because we had observed through measuring optical density, therefore we cannot be able to present it in % of cell death. But for future work, we always consider your suggestion.
Quarry 12. Determination of extracellular enzymes is an important indicator. However, the authors determined their only qualitative availability. It is necessary to quantify the activity and present the data in the results section. It would be useful to determine the content of enzymes, both extracellular and intracellular.

Response. Thanks for suggestion, we have added a figure of extracellular enzyme production data in fig.4, for better and clear presentation

Quarry 13. In the discussion section, give the number of antibiotic-resistant strains, not their %

Response. Thanks for suggestion, we have modified this section in revised manuscript

Round 2

Reviewer 2 Report

remove the capital name and correct the calcium phosphate formula Ca3(PO4)2

the description of the data in table 3 can fit in one sentence

Author Response

Quarry: Remove the capital name and correct the calcium phosphate formula Ca3(PO4)2

Response:  Thanks for this critical suggestion, now corrected in revised manuscript

Quarry: The description of the data in table 3 can fit in one sentence

Response: Removed the table.3 from manuscript